# Current Status and Barriers of Exercise in Individuals with Spinal Cord Injuries in Korea: A Survey

**DOI:** 10.3390/healthcare12101030

**Published:** 2024-05-16

**Authors:** Yuna Kim, Sung-Hwa Ko, Jung-Lim Lee, Sungchul Huh

**Affiliations:** 1Research Institute for Convergence of Biomedical Science and Technology, Pusan National University Yangsan Hospital, Yangsan 50612, Republic of Korea; kyoona1289@gmail.com (Y.K.); ijsh6679@gmail.com (S.-H.K.); lim6668@naver.com (J.-L.L.); 2Department of Rehabilitation Medicine, Pusan National University Yangsan Hospital, Yangsan 50612, Republic of Korea; 3Department of Rehabilitation Medicine, Pusan National University School of Medicine, Yangsan 50612, Republic of Korea

**Keywords:** exercise, spinal cord injury, barrier, community

## Abstract

This study investigated exercise participation, health status, and barriers to exercise in 109 individuals with spinal cord injury (SCI) using a self-report questionnaire. The responses of the exercise and non-exercise groups were statistically analyzed using t-tests or Fisher’s exact test. Significant differences were observed in the cause of injury and the American Spinal Injury Association Impairment Scale between the groups. The non-exercise group had a higher incidence of traumatic and complete injuries. Demographic factors such as gender, age, income level, and marital status did not significantly influence exercise participation. The exercise group reported lower pain scores, less inconvenience from complications, and higher activity and participation scores. However, less than half of the individuals with SCI met the recommended exercise intensity, and community facility usage was low. Barriers to exercise participation included severe disabilities, lack of time, insufficient exercise information, and lack of facility accessibility. To enhance exercise among individuals with SCI, it is essential to develop and extend exercise programs tailored to individual physical factors and a comprehensive understanding of barriers. Prioritizing community-based data management, alongside developing social systems and health policies, is crucial to overcome barriers to exercise participation for individuals with SCI.

## 1. Introduction

Spinal cord injury (SCI) incidence ranges from 40 to 80 cases per one million people worldwide, implying a total of approximately 250,000 to 500,000 individuals affected annually [1]. It is a current trend that the incidence of traumatic SCI is decreasing, while the incidence of non-traumatic SCI, primarily due to degenerative spine diseases, is increasing [2,3,4,5,6,7].

As individuals with SCI are living longer, aging with SCI has become a significant concern in this population [8]. SCI restricts exercise, potentially resulting in increased rates of metabolic syndrome [9]. This significantly increases the mortality rate from cardiovascular disease compared to healthy people [9,10]. Exercise lowers fat mass, lowers low-density lipoprotein cholesterol levels, and helps control blood sugar and inflammation [11]. The higher the level of exercise, the lower the risk of developing chronic diseases and metabolic syndromes [12]. In addition, those who engaged in a sufficient amount of exercise evaluated their pain, stress, and fatigue lower than those who were less physically active, and they reported increased levels of self-efficacy and quality of life [13,14]. Furthermore, high levels of exercise have a positive effect on reducing medical expenses in the long run due to a decrease in the number of hospitalizations and a decrease in dependence on care [15]. Regular exercise was linked to better self-health ratings in SCI individuals [16].

Despite the clear positive effects of exercise, exercise participation rates among individuals with SCI are reported to be significantly lower compared to those without SCI [17,18,19]. The percentage of people with SCI who meet the World Health Organization’s (WHO) recommended amount of exercise ranges from 12% to 48.9% depending on the country, a stark contrast to the 52.8% to 81.8% reported in individuals without SCI [20]. Numerous studies have reported that this disparity is often attributed to external barriers such as the high costs associated with facility use, lack of facilities, limited accessibility, a lack of specialized expertise, and absence of tailored programs [21,22,23,24,25,26,27]. Evidence-based exercise guidelines for people with SCI emphasize the need for a comprehensive understanding of the communities in which they live. This is because when individuals with SCI engage in community exercise, their living environment inevitably influences them [28,29]. Therefore, it is crucial to understand the institutional and cultural characteristics of each country to make appropriate societal adjustments. Several countries, such as the United States, Canada, Australia, and some Asian nations, have established statistical data centers for SCI [30]. However, Korea does not currently have an integrated statistical data center for SCI. This leads to a lack of comprehensive understanding regarding the current status of exercise among this population. To increase the exercise participation rates of people with SCI, it is essential to identify barriers, including Korea’s unique cultural and institutional characteristics, and to understand precisely what support is needed for these individuals to exercise within their communities. To our knowledge, no study has yet investigated and compared how exercise participation and its constraints correlate with health status and quality of life in Korean individuals with SCI.

This study aims to investigate the current status of self-exercise participation among individuals with SCI in Korea and its impact on quality of life, and to identify the barriers preventing participation. We assume that these results can be compared with existing data from other countries and aim to collect data that reflect Korea’s unique cultural and institutional characteristics. This will offer valuable insight into the development of tailored interventions and support systems to enhance exercise participation and the overall wellbeing of people with SCI in Korea.

## 2. Materials and Methods

### 2.1. Survey Design

The cross-sectional survey was conducted using a structured questionnaire and face-to-face interviews with individuals with SCI. The questionnaire was composed of the following sections: (1) personal information, (2) injury characteristics and health status, (3) physical condition and quality of life, (4) activities and participation, and (5) status of self-exercise and needs. In this questionnaire, self-exercise was defined as regular physical activity voluntarily undertaken for health management, excluding any rehabilitation treatment. The questionnaire, which was self-administered, assessed the start time, frequency, intensity, and duration of the exercise, as well as reasons for not exercising. This questionnaire is based on the International Spinal Cord Injury Survey, which was developed by the WHO in collaboration with the International Spinal Cord Injury Society [31]. The full content is available in Appendix A.

Personal information and injury characteristics were collected from medical records, in accordance with the personal information disclosure consent form that was part of the subject description and consent process. Detailed contents of the survey are available in the Appendix A.

### 2.2. Study Sample

The survey was administered to 109 individuals with SCI aged 19 and older at Pusan National University Yangsan Hospital from April to November 2023. Pusan National University Yangsan Hospital is one of the largest government-designated rehabilitation hospitals in Korea, and it is frequented by the greatest number of spinal cord injury patients in the southern part of Korea. All individuals with both traumatic and non-traumatic SCI visiting the outpatient clinic were invited to participate in the survey to minimize selection bias. Exclusion criteria included those who did not comprehend the study details or were unable to respond to the survey, or cases where consent was not given by the participant or their guardian. The study was approved by the Institutional Review Board of Pusan National University Yangsan Hospital (IRB No. 05-2023-053). All participants or their representatives provided written agreement to the subject description and consent form prior to participation. Written informed consent for publication of results is waived, subject to the privacy provisions contained in that consent form.

### 2.3. Statistical Analyses

The data were analyzed using SPSS version 22.0. Participants who engaged in self-exercise within the past three months were categorized into the exercise group, whereas those who had not were placed into the non-exercise group. We compared the demographic characteristics and health statuses between these groups using the t-test or Fisher’s exact test, setting the significance level at 0.05. Results for several multiple-response items are presented as percentages.

## 3. Results

### 3.1. Respondents’ Characteristics

Among the 109 individuals with SCI, 77 (70.64%) were male and 32 (29.36%) were female. Approximately half of the respondents, 50 individuals (45.87%), reported a monthly income of less than 2 million Korean Won. This amount aligns with the average income of the lowest first quintile among the five quintiles of monthly household income for families with two or more people in Korea in 2023 [32]. The types of injury were traumatic in 65 cases (59.63%) and non-traumatic in 44 (40.37%). The most common level of injury was cervical SCI (53.21%), followed by thoracic SCI (36.70%) and lumbar SCI (10.10%). Complete injuries, AIS A with both sensory and motor function below the level of injury impaired, were present in 33 individuals (30.28%), and there were 76 individuals (69.72%) with incomplete injury. Respondents were categorized into two groups based on their responses to the question, “Have you performed self-exercise within the past 3 months?” Those who answered ‘Yes’ formed the exercise group, and those who answered ‘No’ were designated as the non-exercise group. There were 74 individuals (67.89%) in the exercise group, and 35 (32.11%) in the non-exercise group.

### 3.2. Differences in Characteristics Depending on Exercise Performance

No significant differences were observed between the exercise group and the non-exercise group in terms of gender, age, marital status, household composition, income level, time since injury, or neurological level of injury. However, a notable difference was present in the distribution of the American Spinal Injury Association Impairment Scale (AIS) grades. Among the exercisers, 24.32% had a complete injury (AIS A), and 75.68% had an incomplete injury (AIS B-D). In the non-exercisers, 42.86% had a complete injury, and 57.14% had an incomplete injury, as detailed in Table 1. There was a significant difference in injury causation between the two groups. In the exercise group, traumatic and non-traumatic injuries occurred at similar rates. In contrast, the non-exercise group exhibited a traumatic injury rate of 74.29%, marking a notable disparity. The proportion of people with voiding difficulty was 80% in the non-exercise group and 55.41% in the exercise group, showing a significant difference.

### 3.3. Comparison of Individuals’ Status and Quality of Life according to Exercise Participation

Although subjective health status did not differ by exercise level, there was a significant difference in the impact of complications on daily life between the two groups. In the exercise group, 32.43% of the respondents reported that complications were ‘seriously problematic,’ compared to 68.57% in the non-exercise group. Furthermore, the pain score was investigated by presenting an 11-point visual analogue scale to the participants. The exercise group reported lower pain scores, with an average Visual Analog Scale score of 4.82 ± 2.74, in contrast to the non-exercise group’s average score of 6.11 ± 2.27. Significant differences were also observed in the scores for Activities of Daily Living and Mobility, with the exercise group reporting averages of 3.18 ± 1.11 and 3.15 ± 1.19, respectively, compared to the non-exercise group’s averages of 2.57 ± 1.12 and 2.55 ± 1.10. These findings are presented in Table 2.

### 3.4. Status of Community Exercise Facility Usage in Relation to Subjective Health and Injury Severity

Among the respondents, 33 individuals reported using community exercise facilities, while 76 did not. The severity of injury, categorized by the AIS, significantly influenced facility use. Of those with a complete injury (AIS A), 15.15% had used community facilities, in contrast to 46.71% with AIS D, suggesting that facility usage decreased with more severe injuries and increased with milder impairments. Subjective health perception significantly influenced community facility usage, as 28.57% of participants who rated their health status as ‘Good’ had used these facilities, while only 15.38% of those who perceived their health as ‘Very poor’ had experience using the facilities (Table 3).

### 3.5. Exercise Frequency, Intensity, Time, and Place in Exercise Group

Among the 109 participants, 74 (67.89%) reported that they were currently engaged in exercise. Of these, 64.86% began exercising immediately post-discharge, while 18.92% commenced exercise 12 months post-injury. Regular exercise frequency was noted, with 36.49% exercising daily and 40.54% exercising more than three times a week. In terms of intensity, 5.41% engaged in high-intensity work outs, 37.84% in moderate-intensity activities that induced slight shortness of breath, and 54.76% in low-intensity activities like walking and stretching. The average duration of exercise was 80 minutes. Preferred exercise locations varied, with 28 individuals opting for home workouts, 21 attending centers for the disabled, 17 utilizing nearby outdoor spaces like parks and playgrounds, 12 using commercial sports facilities, and 9 choosing public sports amenities (Figure 1).

### 3.6. Barriers to Exercise and Facility Usage

Among the 35 individuals who did not exercise, the most commonly cited reason for not exercising was ‘severe disability’. This was followed by ‘lack of time’, ‘inaccessibility’, and ‘lack of information about exercise’ (Figure 2). Of the 76 individuals surveyed about non-use of community exercise facilities, the most cited reasons were ‘concerns about health status or accidents’ and ‘lack of mobility and accessibility’. When comparing the complete and incomplete injuries, the former reported more difficulties with ‘lack of mobility and accessibility’ (Figure 3). Participants were also queried on the support they require to engage in community exercise. The most common response was the need for ‘an accurate diagnosis of patient condition and determination of exercise intensity’. This was followed by ‘the development of tailored exercise plans and provision of programs by stage’ and ‘provision of exercise-related welfare information and services’ (Figure 4).

## 4. Discussion

For individuals with SCI, participation in exercise is essential for functional improvement, quality of life, and life expectancy [15,33]. To increase the rate of exercise participation, it is crucial to accurately identify the medical characteristics of individuals with SCI and the factors that impede their exercise performance. This study explored whether differences in health and quality of life among people with SCI are associated with their exercise status. We also investigated the barriers to community exercise encountered by individuals.

Our findings underscore the importance of considering an individual’s injury information and health status when facilitating community exercise participation. Injury severity was identified as a significant factor distinguishing between those who exercise and those who do not. Individuals with more severe injuries tended to participate less in exercise activities. However, this study found no significant relationship between exercise participation and gender, age, income level, time since injury, or marital status. The AIS significantly influenced the use of community exercise facilities. For individuals classified as AIS D, the proportions of those with and without experience using these facilities were comparable. However, among those classified as AIS A, only 15% had utilized an exercise facility. Regarding subjective health status, we found that individuals who rated their health positively were more likely to have used such facilities. Nonetheless, subjective health status did not show a significant correlation with exercise performance. This outcome contrasts with previous research, which suggested a direct relationship between higher self-rated health status and increased levels of exercise [16,23]. The exercise and non-exercise groups exhibited significant disparities in terms of the effects of complications in daily life, pain scores, and activity and participation scores. These findings align with the results of previous studies, which suggest that exercise participation reduces pain and enhances the quality of life in individuals with SCI [13,14,33].

The WHO and the evidence-based scientific exercise guidelines for adults with SCI recommend a minimum of 150 minutes of moderate- to high-intensity exercise three times a week [34]. However, our survey revealed that while 67% of participants engaged in exercise, 77.02% did so more than three times a week, with 43.24% performing moderate- to high-intensity exercise. The average exercise duration was 80 minutes, falling short of the guidelines. This rate is comparable to or slightly below the global trend. The proportion of individuals with SCI who meet the recommended exercise guidelines is 48.9% in Switzerland, 44% in the United States, between 12% and 35.5% in Canada, and 28% in the Netherlands [17,18,19,20]. The fact that fewer than half of the participants engage in exercise at the appropriate intensity may be due to a lack of awareness. Alternatively, the challenge may lie in the difficulty of performing high-intensity exercise at home, which is the usual location for their workouts. Therefore, it is posited that educational initiatives on exercise methods and providing accessible venues are essential for enabling individuals with SCI to exercise vigorously enough to become breathless.

‘Severe disability’ was the most frequently cited barrier to exercise, followed by ‘lack of time’, ‘lack of exercise information’, and ‘inadequate facility access’. This demonstrates a marked lack of development and dissemination of differentiated exercise guidelines reflecting the characteristics of individuals with SCI in Korea. To address these obstacles, clinicians and health professionals must develop systematic guidelines for exercise tailored to the needs of individuals with SCI and educate them. On the other hand, none of the participants cited “financial burden” as a barrier to exercise participation. Financial burden also ranked low in barriers that hinder the using community exercise facilities. This contrasts with several previous studies showing that income level has a significant impact on exercise engagement and is considered a key barrier [21,22,23,26]. It should be noted that the perceived difficulties related to income may vary among individuals, even when income levels are quantitatively similar, due to different economic conditions or costs of living in each country. In addition, the low ranking of the item related to internal motivation, ‘lack of necessity’, also shows different results from previous studies. In previous studies, personal internal barriers such as lethargy, perceived difficulty, and lack of interest, rather than economic or social factors, were found to be significant [21,23]. The variation in exercise performance and influencing factors among people with SCI across different countries can be attributed to environmental influences, cultural characteristics, and the social atmosphere that they are exposed to. Based on the findings shown here, individuals with SCI in Korea require more appropriate exercise information and programs rather than motivational education. There is a need to establish and promote more practical and appropriate health and welfare policies, such as supporting transportation for those with reduced mobility.

Our study is significant as it examines the exercise habits of a large cohort of Koreans with SCI, their health status in relation to exercise participation, and the barriers they face. The findings can inform interventions to support these individuals from the initial stages of injury through to societal reintegration, as well as help develop policies to enhance their quality of life. However, some limitations are present. Firstly, since the data are derived from a survey of only 109 individuals in the Gyeongnam region in Korea, they may not reflect the situation across the entire nation. Future research should involve larger samples and expand regionally. Secondly, the self-report nature of the survey could introduce subjective bias, possibly reflecting the respondents’ transient psychological states. This may affect the accuracy and objectivity of the data. Thus, in future studies, it will be necessary to develop and use tools that can objectively measure exercise level and health status, or to increase the reliability of the data through long-term tracking. Lastly, the time elapsed since injury was not considered. Considering that SCI recovery involves various psychological stages from the time of injury to status acceptance, it is crucial to distinguish between acute and chronic injuries in relation to the injury period. Subsequent studies should aim to objectively measure and evaluate depression, anxiety, and anger, and involve larger samples from multiple institutions for comprehensive analysis.

## 5. Conclusions

This study investigated the exercise behaviors of individuals with SCI and examined how health factors impact their exercise and use of exercise facilities. Given that the health status and quality of life for those with SCI are closely linked to their level of exercise, national efforts to promote exercise are essential. To facilitate this, there is a need to create and implement exercise programs that incorporate precise diagnostics reflecting the health factors identified in this study. Furthermore, community-based health status evaluations and environmental management for individuals with SCI are crucial. Based on these insights, social institutions and health policies must be developed to address and eliminate barriers to exercise participation.

## Figures and Tables

**Figure 1 healthcare-12-01030-f001:**
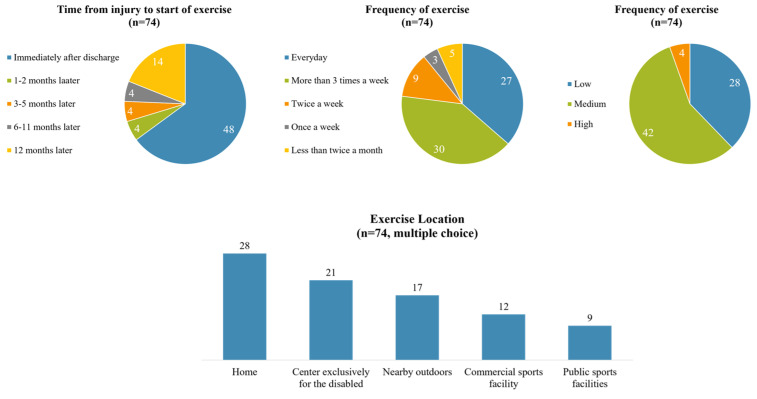
Current exercise status in exercise group.

**Figure 2 healthcare-12-01030-f002:**
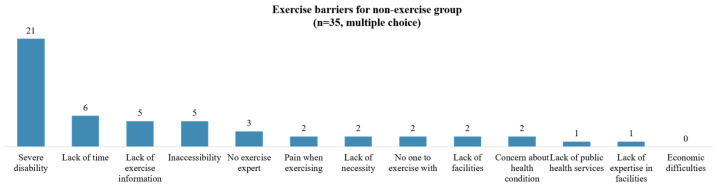
Exercise barriers for non-exercise group.

**Figure 3 healthcare-12-01030-f003:**
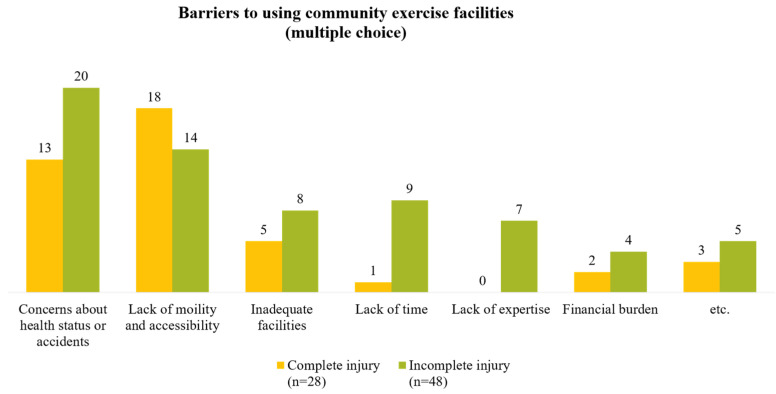
Reasons for not using community exercise facilities among individuals with complete and incomplete injuries.

**Figure 4 healthcare-12-01030-f004:**
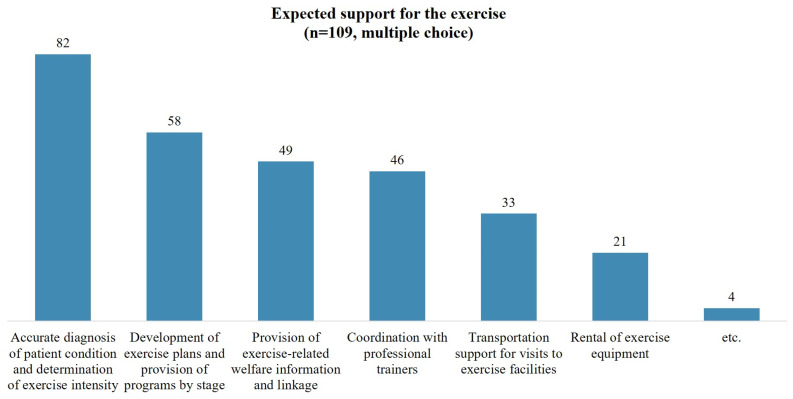
Expected support for the exercise.

**Table 1 healthcare-12-01030-t001:** Differences in characteristics depending on exercise performance.

Variables	Categories	Exercise Group *n* (%)	Non-Exercise Group *n* (%)	*p*-Value
Gender	Male	49 (66.22)	28 (80.00)	0.140
	Female	25 (33.78)	7 (20.00)	
Age (years)	19~28	4 (5.41)	0 (0.0)	0.134
	29~38	0 (0.0)	1 (2.86)	
	39~48	11 (14.86)	6 (17.14)	
	49~58	19 (25.68)	15 (42.86)	
	59~68	27 (36.49)	7 (20.00)	
	Over 69	13 (17.57)	6 (17.14)	
Marital status	Single	13 (17.57)	9 (25.71)	0.375
	Married	52 (70.27)	21 (60.00)	
	Cohabiting	1 (1.35)	0 (0.0)	
	Divorced or separated	5 (6.76)	5 (14.29)	
	Widowed	3 (4.05)	0 (0.0)	
Household composition	Live alone	14 (18.92)	10 (28.57)	0.375
	Live with more than two people	60 (81.08)	25 (71.43)	
Income level (Million Korean Won)	Under 2	29 (39.19)	21 (60.00)	0.216
	2~3.99	24 (32.43)	5 (14.29)	
	4~5.99	10 (13.51)	4 (11.43)	
	5~7.99	8 (10.81)	5 (14.29)	
	8~9.99	2 (2.70)	0 (0.0)	
	Over 10	1 (1.35)	0 (0.0)	
Time since injruy (years)	Under 0.5	1 (1.35)	0 (0.0)	0.346
	0.5~1	4 (5.41)	2 (5.71)	
	1~5	26 (35.14)	6 (17.14)	
	5~10	15 (20.27)	10 (28.57)	
	Over 10	28 (37.84	17 (48.57)	
Type of injury	Traumatic	39 (52.70)	26 (74.29)	0.032 *
	Non-traumatic	35 (47.30)	9 (25.71)	
Level of injury	Cervical	38 (51.35)	20 (57.14)	0.831
	Thoracic	28 (37.84)	12 (34.29)	
	Lumbar	8 (10.81)	3 (8.57)	
	Sacral	0 (0.0)	0 (0.0)	
ASIA ^1^ Impairment Scale	A	18 (24.32)	15 (42.86)	0.018 *
	B	6 (8.11)	6 (17.14)	
	C	11 (14.86)	6 (17.14)	
	D	39 (52.70)	8 (22.86)	
Smoking	Yes	12 (16.22)	7 (20.0)	0.627
	No	62 (83.78)	28 (80.0)	
Drinking	Yes	14 (18.92)	6 (17.14)	0.823
	No	60 (81.08)	29 (82.86)	
Voiding difficulty	Yes	41 (55.41)	28 (80.0)	0.013 *
	No	33 (44.59)	7 (20.0)	
Defecation difficulty	Yes	44 (59.46)	27 (77.14)	0.070
	No	30 (40.54)	8 (22.86)	

^1^ ASIA; American Spinal Injury Association. * *p* < 0.05.

**Table 2 healthcare-12-01030-t002:** Comparison of individuals’ status and quality of life according to exercise participation.

		Exercise Group (*n* = 74)	Non-Exercise Group (*n* = 35)	*p*-Value
Impact of complication on daily life	No problem at all	6 (8.11)	1 (2.86)	0.012 *
	Slightly problematic	9 (12.16)	1 (2.86)	
	Average	9 (12.16)	2 (5.71)	
	Slightly problematic	26 (35.14)	7 (20.00)	
	Severely problematic	24 (32.43)	24 (68.57)	
Subjective health perception	Very good	0 (0.0)	0 (0.0)	0.329
	Good	4 (5.4)	3 (8.6)	
	Fair	38 (51.4)	12 (34.3)	
	Poor	25 (33.8)	14 (40.0)	
	Very poor	7 (9.5)	6 (17.1)	
Pain score		4.82 ± 2.74	6.11 ± 2.27	0.017 *
Score for activity	Activities of Daily Living	3.18 ± 1.11	2.57 ± 1.12	0.009 *
	Mobility	3.15 ± 1.19	2.55 ± 1.10	0.013 *
	Function and participation	3.20 ± 1.08	2.78 ± 0.98	0.055

Values are given as mean ± standard deviation and median [range]. * *p* < 0.05.

**Table 3 healthcare-12-01030-t003:** Status of community exercise facility usage in relation to subjective health and injury severity.

Experience with Community Exercise Facilities	Subjective Health Perception					ASIA ^1^ Impairment Scale				
	Good	Fair	Poor	Very Poor	*p*-Value	A	B	C	D	*p*-Value
Yes *n*, (%)	2 (28.57)	22 (44.00)	2 (17.95)	2 (15.38)	0.033 *	5 (15.15)	2 (16.67)	4 (23.53)	22 (46.71)	0.013 *
No *n*, (%)	5 (71.43)	28 (56.00)	32 (82.05)	11 (84.62)		28 (84.85)	10 (83.33)	13 (76.47)	25 (53.19)	

^1^ ASIA; American Spinal Injury Association. * *p* < 0.05.

## Data Availability

The dataset analyzed in the present study is not publicly available because of ethical and legal regulations regarding the protection of personal data.

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
