# Peer review of "Current Status and Barriers of Exercise in Individuals with Spinal Cord Injuries in Korea: A Survey"

_healthcare, 2024, doi:10.3390/healthcare12101030_

Round 1

Reviewer 1 Report

Comments and Suggestions for Authors

In this manuscript, Kim et al. perform a survey of individuals with spinal cord injury attending an outpatient clinic in Korea to evaluate the rates of physical activity and exercise along with potential barriers to performing exercise. The cultural and environmental status of individuals with SCI can have a large influence on patient outcomes, and there is no Korean statistical database of SCI, as there is in other countries. Therefore, there is a distinct need to evaluate exercise use and frequency in the Korean SCI population to develop strategies to improve health and wellness in this population. The data and text here is presented effectively. However, my main concern is that there are potentially relevant variables that were asked in the survey that are not included as results. This along with other issues are discussed in detail below.

Major Comments:

·       The last introductory paragraph should end with a clear hypothesis/goal that the results can be compared against.

·       The title/abstract should emphasize that the study was focused on Korean individuals. As stated in the introduction, the unique cultural and institutional barriers experienced by the respondents may not translate to other locations.

·       Information on the time since injury was collected according to the survey, but that information does not appear to be included in the results? This could be an informative result as those with more acute SCI may be more motivated to exercise compared to those with longer-term injuries. Therefore, developing strategies to encourage exercise based on the chronicity of SCI may be necessary.

·       There are other elements that were obtained through the survey that could also play an important role that do not appear to be reported (e.g., smoking, drinking, urination, defecation, self-care)? Although these factors may not directly influence an individuals’ decision to exercise, they contribute to the overall self-image of the individual and increased drug use or lower confidence in bladder/bowel function may interfere with seeking out community exercise programs. All of this data should be reported even if it is statistically non-significant.

·       What are the demographics of SCI patients who typically attend the outpatient clinic at Pusan National University Yangsan Hospital? Are they mainly rural or urban? Results from Canadian surveys of individuals with SCI have indicated differences in environmental barriers to care and decreased infrastructure in rural areas can affect patient outcomes (e.g., Glennie et al J Neurotrauma 2017).

·       The introduction indicates that there are major statistical centers for SCI in the US, Canada, Australia, and non-Korean Asian nations. Do those centers include information on physical activity/exercise? It would be interesting to include a more in-depth comparison to see if there are similarities or differences in activity across cultures. It may also be useful to compare these results against recent demographic studies of SCI in Korea (Kim et al. Ann Rehabil Med 2023)..

Minor Comments:

·       Page 1, Lines 32-33: “It is a current trend that the incidence of traumatic SCI is decreasing, while the incidence of non-traumatic SCI, primarily due to degenerative spine diseases, is increasing.” This is referencing a paper solely examining this statistic in Korea. Do these trends hold worldwide?

·       Reference 3 is for a sentence focused on aging in SCI, but the reference is to a study focused on functional outcomes without a specific focus on aging. There are more appropriate citations to use for this sentence (e.g., Groah SL, et al. Am J Phys Med Rehabil. 2012).

·       Page 1, Lines 35-36: “SCI restricts physical activity and exercise, potentially resulting in increased rates of metabolic syndrome.” Page 1, Lines 39-40: “The higher the level of physical activity and exercise, the lower the risk of developing chronic diseases and metabolic syndromes.” There are no citations given to confirm this information.

·       Page 1, Lines 40-41: “In addition, it reduces pain, stress, fatigue, and improves self-efficacy, quality of life.” Here, “it” refers to physical activity? Or exercise? Or both? In general, use of the word “it” should be minimized in scientific writing if possible as it can be unclear what is being referred to. The following sentence exhibits the same issue.

·       Page 2, Line 45: typo - “of among”, only one of these words is needed. This same typo “in among” is made in line 49.

·       Page 2, Lines 55-57: “Several countries, such as the United States, Canada, Australia, and some Asian nations, have established statistical data centers for SCI.” These data centers should be specifically referenced.

·       Page 3, Lines 107-108: “Complete injuries were present in 33 individuals (30.28%), and there were 76 individuals (30.28%) with incomplete injury.” What does “complete” here mean? ASIA-A? Also, the two percentages are identical.

·       Page 3, Line 113: “No significant differences were observed between the two groups in terms of…” The two groups being exercisers and non-exercisers?

·       Page 4, Line 130: “…with an average Visual Analog Scale score of…” Were the participants presented with a Visual Analog Scale or were they simply asked to rate their pain on a scale of 1-10?

·       Figure 1 has a typo: “Nerby outdore” should be “Nearby outdoors”

·       What does the “etc.” category indicate in Figure 2? Only 1 response is given, so it seems strange not to just state what the response was?

·       Figure 3 caption: “Reasons for not use of…” should be “Reasons for not using…”

·       In Figures 1 and 2, does the red bar simply indicate the maximum value? Or is there a reason why the color is different?

·       Page 7, Line 200: “laziness” is a bit derogatory. Perhaps “lethargy” would be superior?

·       Page 8, Line 233: typo “Korean” -> “Koreans”

·       Page 8, Line 247: “…and involving” should be “and involve”

·       The Appendix A listing at the end of the manuscript is the generic entry for a MDPI paper. This should be edited to include the actual information in the appendix prior to publication.

Comments on the Quality of English Language

See minor comments.

Author Response

We thank you and the reviewers for your thoughtful suggestions and insights. The manuscript has benefited from these insightful suggestions.

  1. The last introductory paragraph should end with a clear hypothesis/goal that the results can be compared against.

Response: (Lines 71-77) Thank you for the insightful point. I concur with your opinion and have revised the paragraph accordingly. We have articulated a clear hypothesis and goal.

  1. The title/abstract should emphasize that the study was focused on Korean individuals. As stated in the introduction, the unique cultural and institutional barriers experienced by the respondents may not translate to other locations.

Response: (Lines 2-3) I completely agree with your comment. We have changed the title of the article to “Current status and barriers of physical activity and exercise in individuals with spinal cord injuries in Korea: a survey”.

  1. Information on the time since injury was collected according to the survey, but that information does not appear to be included in the results? This could be an informative result as those with more acute SCI may be more motivated to exercise compared to those with longer-term injuries. Therefore, developing strategies to encourage exercise based on the chronicity of SCI may be necessary.

Response: (Line 127, Table 1) Taking your comments into consideration, we investigated the statistical relationship between the period after injury and exercise performance. The results yielded a p-value of 0.346, indicating no statistical significance. Believing this information to be of importance, we have included it in the main text.

  1. There are other elements that were obtained through the survey that could also play an important role that do not appear to be reported (e.g., smoking, drinking, urination, defecation, self-care)? Although these factors may not directly influence an individuals’ decision to exercise, they contribute to the overall self-image of the individual and increased drug use or lower confidence in bladder/bowel function may interfere with seeking out community exercise programs. All of this data should be reported even if it is statistically non-significant.

Response: I agree with your opinion. We have added other elements (such as smoking, drinking, urination, and defecation) to the survey in Table 1.

  1. What are the demographics of SCI patients who typically attend the outpatient clinic at Pusan National University Yangsan Hospital? Are they mainly rural or urban? Results from Canadian surveys of individuals with SCI have indicated differences in environmental barriers to care and decreased infrastructure in rural areas can affect patient outcomes (e.g., Glennie et al J Neurotrauma 2017).

Response: Pusan National University Yangsan Hospital is one of the largest government-designated rehabilitation hospital in Korea. Although not located in the metropolitan area, it is frequented by the greatest number of spinal cord injury patients in the southern part of Korea. Busan (Pusan) is second largest city in Korea.

  1. The introduction indicates that there are major statistical centers for SCI in the US, Canada, Australia, and non-Korean Asian nations. Do those centers include information on physical activity/exercise? It would be interesting to include a more in-depth comparison to see if there are similarities or differences in activity across cultures. It may also be useful to compare these results against recent demographic studies of SCI in Korea (Kim et al. Ann Rehabil Med2023).

Response: (Lines 237-259) The centers do not have information on physical activities and exercises. However, I believe that nationally managing data on the lives of people with spinal cord injury will be very helpful in promoting physical activity and exercise. A comparison of cross-cultural movement participation status and major barriers has been added to the Discussion.

  1. Page 1, Lines 32-33: “It is a current trend that the incidence of traumatic SCI is decreasing, while the incidence of non-traumatic SCI, primarily due to degenerative spine diseases, is increasing.” This is referencing a paper solely examining this statistic in Korea. Do these trends hold worldwide?

Response: (Lines 32-34, Reference 3-7) I have added several references that refer to the current trend of increased non-traumatic SCI.

  1. Reference 3 is for a sentence focused on aging in SCI, but the reference is to a study focused on functional outcomes without a specific focus on aging. There are more appropriate citations to use for this sentence (e.g., Groah SL, et al. Am J Phys Med Rehabil. 2012).

Response: (Line 36, Reference 8) Thank you for the valuable point. The reference has been updated to include relevant paper that focus on aging.

  1. Page 1, Lines 35-36: “SCI restricts physical activity and exercise, potentially resulting in increased rates of metabolic syndrome.” Page 1, Lines 39-40: “The higher the level of physical activity and exercise, the lower the risk of developing chronic diseases and metabolic syndromes.” There are no citations given to confirm this information.

Response: (Lines 36-37 and Lines 40-41) We have added appropriate references for the sentences.

  1. Page 1, Lines 40-41: “In addition, it reduces pain, stress, fatigue, and improves self-efficacy, quality of life.” Here, “it” refers to physical activity? Or exercise? Or both? In general, use of the word “it” should be minimized in scientific writing if possible as it can be unclear what is being referred to. The following sentence exhibits the same issue.

Response: (Lines 42-44 and Lines 44-45) I agree with your comment. The sentences have been edited for enhanced clarity.

  1. Page 2, Line 45: typo - “of among”, only one of these words is needed. This same typo “in among” is made in line 49.

Response: (Lines 49 and 53) We have corrected the grammatical errors.

  1. Page 2, Lines 55-57: “Several countries, such as the United States, Canada, Australia, and some Asian nations, have established statistical data centers for SCI.” These data centers should be specifically referenced.

Response: We simply intended to note that there are data centers in other countries, but none in Korea. We did not wish to specify what their data are.

  1. Page 3, Lines 107-108: “Complete injuries were present in 33 individuals (30.28%), and there were 76 individuals (30.28%) with incomplete injury.” What does “complete” here mean? ASIA-A? Also, the two percentages are identical.

Response: (Lines 118-120) A complete injury is indicated by ASIA Impairment Scale A, we have revised the terminology and updated the percentages for both complete and incomplete injuries.

  1. Page 3, Line 113: “No significant differences were observed between the two groups in terms of…” The two groups being exercisers and non-exercisers?

Response: (Line 125-126) The sentence has been modified to avoid ambiguity.

  1. Page 4, Line 130: “…with an average Visual Analog Scale score of…” Were the participants presented with a Visual Analog Scale or were they simply asked to rate their pain on a scale of 1-10?

Response: (Lines 141-143) Participants were provided with an 11-point visual analogue scale. An explanatory sentence has been added.

  1. Figure 1 has a typo: “Nerby outdore” should be “Nearby outdoors”

Response: (Figure 1) Thank you for correcting the alphabetical error.

  1. What does the “etc.” category indicate in Figure 2? Only 1 response is given, so it seems strange not to just state what the response was?

Response: (Figure 2) I completely agree with your opinion. “etc.” was deleted and rephrased.

  1. Figure 3 caption: “Reasons for not use of…” should be “Reasons for not using…”

Response: (Figure 3) We have changed the sentence as your comment. “Reasons for not using community exercise facilities among individuals with complete and incomplete injuries.”

  1. In Figures 1 and 2, does the red bar simply indicate the maximum value? Or is there a reason why the color is different?

Response: (Figure 1 and Figure 2) Yes, to indicate the maximum value. However, it was determined that differentiation was not essential; therefore, the colors were unified.

  1. Page 7, Line 200: “laziness” is a bit derogatory. Perhaps “lethargy” would be superior?

Response: (Line 251) I agree with your opinion and have revised the wording accordingly.

  1. Page 8, Line 233: typo “Korean” à “Koreans”

Response: (Line 260) We have changed the word to ‘Koreans’.

  1. Page 8, Line 247: “…and involving” should be “and involve”

Response: (Line 274) We have corrected the grammatical errors.

  1. The Appendix A listing at the end of the manuscript is the generic entry for a MDPI paper. This should be edited to include the actual information in the appendix prior to publication.

Response: (Lines 304-305) The Appendix A has been modified describing the actual information.

Thank you.

Reviewer 2 Report

Comments and Suggestions for Authors

"Current status and barriers of physical activity and exercise in individuals with spinal cord injuries: a survey".

 Strengths of the Article:

The study examined the exercise behaviors of individuals with spinal cord injury (SCI) and assessed how health factors influence their physical activity levels and utilization of exercise facilities. Given the close connection between the health status and quality of life of individuals with SCI and their engagement in physical activity, it is imperative to undertake national initiatives to promote exercise. To achieve this goal, there is a pressing need to develop and implement exercise programs incorporating precise diagnostics reflecting the health factors identified in this study. Additionally, community-based management of health status and environmental factors for individuals with SCI is paramount. Drawing from these insights, it is essential to establish social institutions and health policies to identify and eliminate barriers to exercise participation.

While the study offers valuable insights, several critical refinements are necessary to enhance clarity and improve the manuscript's readability. The major critiques and suggestions are outlined below:

Major Critiques and Suggestions:

1. Is there a significant correlation between injury severity or time elapsed since injury and the level of physical activity and exercise engagement?

2. The survey encompassed 109 individuals with SCI from Pusan National University Yangsan Hospital over a relatively short period from April to November 2023, raising concerns about the representation of the findings at a national level. How accurately do these findings reflect the situation across the entire country?

3. It is advisable to include a dedicated section on study limitations to foster critical engagement with methodological challenges and potential constraints on the generalizability of results.

 4. Given that a significant proportion of participants in the survey had cervical SCI (53.21%), with a presumption that all 21 individuals with severe disabilities belonged to this group in Figure 2, clarification is needed regarding the etiology of these SCIs—whether traumatic or non-traumatic.

 5. Figures 1 and 3 suffer from poor readability of the text. Improving the quality of these figures is recommended.

Comments on the Quality of English Language

Moderate editing of the English language and formatting is required. 

Author Response

We thank you and the reviewers for your thoughtful suggestions and insights. The manuscript has benefited from these insightful suggestions. 

  1. Is there a significant correlation between injury severity or time elapsed since injury and the level of physical activity and exercise engagement?

Response: (Line 127, Table 1) These contents (injury severity, time elapsed since injury) have been added to the results section. However, there was no statistically significant relationship.

  1. The survey encompassed 109 individuals with SCI from Pusan National University Yangsan Hospital over a relatively short period from April to November 2023, raising concerns about the representation of the findings at a national level. How accurately do these findings reflect the situation across the entire country?

Response: Pusan National University Yangsan Hospital is one of the largest government-designated rehabilitation hospital in Korea. Although not located in the metropolitan area, it is frequented by the greatest number of spinal cord injury patients in the southern part of Korea. Busan (Pusan) is second largest city in Korea.

  1. It is advisable to include a dedicated section on study limitations to foster critical engagement with methodological challenges and potential constraints on the generalizability of results.

Response: (Lines 264-267) I have addressed the methodological challenge of selection bias, which could lead to generalizability deficits, given that our study data were collected only from the southern part of Korea.

  1. Given that a significant proportion of participants in the survey had cervical SCI (53.21%), with a presumption that all 21 individuals with severe disabilities belonged to this group in Figure 2, clarification is needed regarding the etiology of these SCIs—whether traumatic or non-traumatic.

Response: Thank you for your thoughtful comment. However, we believe that barriers to the exercise participation depends more on the severity of the disability than on the etiology of the SCI, and thus dividing the etiology for understanding the barriers is not required.

  1. Figures 1 and 3 suffer from poor readability of the text. Improving the quality of these figures is recommended.

Response: (Figure 1 and Figure 3) Thank you for your comment. The image quality has been improved.

Thank you.

Round 2

Reviewer 1 Report

Comments and Suggestions for Authors

Thank you to the authors for sufficiently responding to my previous comments. I have a few minor follow-up comments that should be addressed, but the manuscript is overall much improved.

·       The values for smoking, drinking, urination, and defecation were included in Table 1, but the p-value should also be included as it is for the other categories.

·       The answer regarding the demographics of Pusan National University Yangsan Hospital is appropriate, but this information should also be reflected in an update to the text to the manuscript.

·       Lines 42-44: “In addition, those who engaged a sufficient amount of physical activity and exercise evaluated their pain, stress, fatigue lower than those who were less physical activity and exercise, and they reported self-efficacy and quality of life more highly” This updated sentence has some grammatical issues. “In addition, those who engaged in a sufficient amount of physical activity and exercise evaluated their pain, stress, and fatigue lower than those who were less physically active, and they reported increased levels of self-efficacy and quality of life.” is better grammatically.

·       The bar charts are an improvement over the previous figure visualizations. Minor comment in the new Figure 4 and legend, “supports” should be singular “support”

·       Line 243-244: “none of participants” -> “none of the participants”, and “It” in the subsequent sentence should be replaced by “Financial burden”.

·       Lines 255-257: The sentence starting with “Currently, individuals…” could be improved to “Based on the findings shown here, individuals with SCI in Korea require more appropriate exercise information and programs rather than motivational education.”

Author Response

We thank you for your thoughtful suggestions.

  1. The values for smoking, drinking, urination, and defecation were included in Table 1, but the p-value should also be included as it is for the other categories.

Response: (Table 1) Thanks for the advice. All p-values have been added. Additionally, the Yes/No category results for the urination and defecation items in Table 1 were written in reversed rows, so they were corrected. The proportion of people with voiding difficulty was 80% in the non-exercise group and 55.41% in the exercise group. And the terms were modified more appropriately from “urination disorder” and “defecation disorder” to “voiding difficulty” and “defecation difficulty”.

  1. The answer regarding the demographics of Pusan National University Yangsan Hospital is appropriate, but this information should also be reflected in an update to the text to the manuscript.

Response: (Lines 93-96) I agree with you. I added it.

  1. Lines 42-44: “In addition, those who engaged a sufficient amount of physical activity and exercise evaluated their pain, stress, fatigue lower than those who were less physical activity and exercise, and they reported self-efficacy and quality of life more highly” This updated sentence has some grammatical issues. “In addition, those who engaged in a sufficient amount of physical activity and exercise evaluated their pain, stress, and fatigue lower than those who were less physically active, and they reported increased levels of self-efficacy and quality of life.” is better grammatically.

Response: (Lines 42-44) Thank you for reporting the grammatical error. I changed it to the sentence you recommended.

  1. The bar charts are an improvement over the previous figure visualizations. Minor comment in the new Figure 4 and legend, “supports” should be singular “support”

Response: (Figure 4) Grammar errors have been corrected.

  1. Line 243-244: “none of participants” -> “none of the participants”, and “It” in the subsequent sentence should be replaced by “Financial burden”.

Response: (Lines 245-247) Grammar errors have been corrected.

  1. Lines 255-257: The sentence starting with “Currently, individuals…” could be improved to “Based on the findings shown here, individuals with SCI in Korea require more appropriate exercise information and programs rather than motivational education.”

Response: (Lines 259-260) I think the sentence you provided is much more appropriate. It has been modified. Thank you
